# Oxidation of Ethanol in Cu-Faujasites Studied by IR Spectroscopy

**DOI:** 10.3390/molecules26092669

**Published:** 2021-05-02

**Authors:** Łukasz Kuterasiński, Jerzy Podobiński, Jerzy Datka

**Affiliations:** Jerzy Haber Institute of Catalysis and Surface Chemistry, Polish Academy of Sciences, ul. Niezapominajek 8, 30-239 Kraków, Poland; nckutera@cyf-kr.edu.pl (Ł.K.); ncpodobi@cyf-kr.edu.pl (J.P.)

**Keywords:** ethanol oxidation, Cu species, faujasite

## Abstract

In this study, IR studies of the coadsorption of ethanol and CO on Cu^+^ cations evidenced the transfer of electrons from ethanol to Cu^+^, which caused the lowering of the frequency of the band attributed to CO bonded to the same Cu^+^ cation due to the more effective π back donation of d electrons of Cu to antibonding π* orbitals of CO. The reaction of ethanol with acid sites in zeolite HFAU above 370 K produced water and ethane, polymerizing to polyethylene. Ethanol adsorbed on zeolite Cu(2)HFAU containing acid sites and Cu^+^_exch_ also produced ethene, but in this case, the ethene was bonded to Cu^+^ and did not polymerize. C=C stretching, which is IR non-active in the free ethene molecule, became IR active, and a weak IR band at 1538 cm^−1^ was present. The reaction of ethanol above 370 K in Cu(5)NaFAU zeolite (containing small amounts of Cu^+^_exch_ and bigger amounts of Cu^+^_ox_, Cu^2+^_exch_ and CuO) produced acetaldehyde, which was further oxidized to the acetate species (CH_3_COO^−^). As oxygen was not supplied, the donors of oxygen were the Cu species present in our zeolite. The CO and NO adsorption experiments performed in Cu-zeolite before and after ethanol reaction evidenced that both Cu^+^_ox_ and Cu^2+^ (Cu^2+^_exch_ and CuO) were consumed by the ethanol oxidation reaction. The studies of the considered reaction of bulk CuO and Cu_2_O as well as zeolites, in which the contribution of Cu^+^_ox_ species was reduced by various treatments, suggest that ethanol was oxidized to acetaldehyde by Cu^2+^_ox_ (the role of Cu^+^_ox_ could not be elucidated), but Cu^+^_ox_ was the oxygen donor in the acetate formation.

## 1. Introduction

Cu-containing zeolites gained a great deal of attention because of their activity in the decomposition or reduction of nitrogen oxides [1,2,3,4,5,6,7,8,9,10,11,12,13], the ability of activation of multiple bonds in organic molecules [14,15,16,17,18,19,20] and the catalytic activity in various organic molecules. Alcohol oxidation is one of such reactions catalyzed by copper containing zeolites [21,22,23,24,25,26,27,28,29,30,31,32,33,34,35,36,37]. For instance, Tsuruva et al. [21] investigated the vapor-phase oxidation of benzyl alcohol over CuNa-Y zeolite in the temperature range of 573 to 663 K. The main products were CO, CO_2_, and benzaldehyde. Cu(II) ion was found as the main active site for this reaction. The benzaldehyde formation rate was well described by the following equation:r_φCHO_ = k · [O_2_]^1^[φCH_2_OH]^−1^) 
with the activation energy of 57.0 kJ/mol. Furthermore, it was indicated that the catalytic activity for the oxidation of benzyl alcohol depended on the addition of amine to reaction system. The addition of piperidine decreased, and the addition of pyridine increased, the oxidation activity. From Electron Spin Resonance (ESR) measurements of Cu(II)NaY-amine systems, it was evidenced that the strength of the covalent bonding between Cu(II) and piperidine was stronger in comparison with pyridine. It was also found that the addition of CO and H_2_ into the reaction system decreased the conversion of benzyl alcohol and changed the selectivity to benzaldehyde, which was higher for hydrogen, on the other hand, a water-deactivated catalyst.

In another work, Tsuruva et al. [22] reported the gas-phase oxidation of ethanol over CuNa-Y at 523–623 K. Acetaldehyde was formed as the main oxidation product in the whole temperature range. It was also evidenced that Cu(II) ions were the active sites responsible for the oxidative dehydrogenation of ethanol. Kinetic studies of acetaldehyde formation over the CuNa-Y catalyst evidenced that the acetaldehyde formation rate was reciprocal, first order in ethanol and one-half order in relation to the partial pressure of oxygen, according to the Equation (1), with the activation energy of 70.0 kJ/mol.
r_CH3CHO_ = kP^−1^_C2H5OH_P^1/2^_O2_(1)

Bun et al. [23] investigated ZSM-5 type zeolites containing protons or Cu^II^ as catalysts for the ethanol conversion under the atmosphere, both in the presence and absence of oxygen. Analysis of the collected data led to the conclusion that the protonic acid sites present in ZSM-5 favored the dehydration of ethanol toward ethene, irrespective of the presence of oxygen in the reaction system. In turn, Cu(II) ion in the ZSM-5 promoted the oxidation of ethanol, which led to the production of CO and CO_2_.

Alcohol can be used not only as a precursor, but also as a reducing agent in the selective catalytic reduction of nitrogen oxides. The influence of the copper content (0.3, 1.5 and 3.3 wt.%) on the catalytic properties of Cu_x_SiBEA zeolites in SCR of NO by ethanol was investigated by Janas et al. [38]. According to XRD, DR UV–vis–NIR and XPS results, copper incorporated into the dealuminated SiBEA zeolite was present mainly in the form of isolated tetracoordinated Cu(II). Cu_0.3_SiBEA, Cu_1.5_SiBEA and Cu_3.3_SiBEA were active in SCR, for which NO conversions at 573 K were 33%, 45.5% and 50%, whereas the selectivity to N_2_ was 90%, 97% and 75%, respectively. Basing on these results, it was concluded that the SCR of NO took place on isolated tetracoordinated Cu^2+^ in the absence of Al atoms. Conversion of NO increased with copper content, while selectivity to N_2_ was independent of the amount of Cu.

In our earlier paper [39], we reported that the way of preparation of copper containing faujasite-type zeolite of Si/Al = 31 determined the status and properties of Cu sites in these catalysts. Cu was found in the form of monovalent cation occurring both in exchange extra-framework positions (Cu^+^_exch_) and as oxides (Cu^+^_ox_), as well as in the form of divalent species (Cu^2+^_exch_ and CuO). Cu cations in the exchange extra framework positions are ions, which neutralize the negative charge of AlO_4_^−^ tetrahedral. The cations in oxide forms are neutralized by the negative charge of the framework, which is neutralized by protons or exchange cations. The amounts of Cu forms depended on the copper content in zeolites and on the form of zeolite support of the faujasite structure of Si/Al = 31 to which Cu was incorporated: commercial dealuminated faujasite in protonic form (HFAU) or faujasite after ion-exchange with NaNO_3_ (NaFAU). The former type of zeolites (CuHFAU) contained mainly Cu^+^_exch_, while for CuNaFAU-type zeolites, much higher amounts of Cu^+^_ox_ and Cu^2+^ were found. Furthermore, it was indicated that both Cu^+^ and Cu^2+^ in oxide forms showed stronger electrodonor properties in comparison with Cu^+^ and Cu^2+^ in the exchange form.

We also reported the process of reduction and oxidation of the Cu species in CuFAU [40]. In our study, reducers were hydrogen and ethanol, while the role oxidizers played were oxygen and NO. In the reduction experiments, it was revealed that Cu^+^_ox_ and Cu^2+^_exch_ were more prone to reduction by hydrogen than Cu^+^_exch_ and Cu^2+^_ox_ Ethanol caused reduction mainly for Cu^2+^ and partially for Cu^+^.

The present study concerns the reaction of ethanol with the Cu species in Cu-zeolites. Two zeolites were studied: Cu(2)HFAU and Cu(5)NaFAU. These zeolites contained 2 and 5 wt.% of Cu, respectively. Copper was introduced by impregnation to the protonic (HFAU) or sodium form of faujasite (NaFAU), respectively. Zeolite Cu(2)HFAU contained mostly Cu^+^_exch_, whereas in Cu(5)NaFAU, relatively large amounts of both Cu^+^_ox_ and Cu^2+^_,_ and smaller Cu^+^_exch_ contents were found. That allowed us to determine the ability of various Cu species (i.e., copper in exchange positions as well as Cu in the form of oxides) in the oxidation of ethanol. We also studied the coadsorption of ethanol and CO molecules on Cu^+^ cations. This informed how the interaction of Cu^+^ with ethanol changes the electronic properties of Cu^+^.

## 2. Results and Discussion

### 2.1. Coadsorption of Ethanol and CO

In order to learn how the interaction of Cu^+^ ions with ethanol molecules modifies the electronic properties of Cu^+^, we performed the coadsorption of ethanol and CO molecules over Cu(2)HFAU. Sorption of ethanol at room temperature was followed by the sorption of CO on the studied catalyst at the same temperature. The spectrum of CO interacting with Cu^+^_exch_ showed the band at 2158 cm^−1^, which was also observed in our previous studies [39,40] and was reported by other authors [3,41,42,43]. When ethanol was preadsorbed, the Cu^+^_exch_–CO band was shifted to lower frequencies and the two shifted maxima appeared at 2118 and 2134 cm^−1^ (Figure 1). Lowering the CO stretching frequency is the result of electron transfer from the ethanol molecule to Cu^+^ ion and the more efficient effect of π-back donation of d-electrons of Cu^+^ to π^*^ antibonding orbitals of CO. Similar effects were observed in earlier studies of coadsorption of other molecules, such as but-1-ene, benzene, acetone and CO on Cu^+^ in CuZSM-5 [44]. Two bands assigned to Cu^+^_exch_ interacting simultaneously with CO and ethanol molecules (2118 and 2134 cm^−1^) suggested the occurrence of two kinds of Cu^+^_exch_ ions of different electrondonor properties. It may be the result of the presence of two kinds of Cu^+^_exch_ modified to various extents by electron transfer from ethanol molecules.

### 2.2. Reaction of Ethanol in HFAU

As Cu-zeolites contain not only Cu sites but also Brønsted acid sites (Si–OH–Al groups, Figure 2A), we studied the transformations of ethanol sorbed in zeolite HFAU containing only protonic sites. The spectra recorded upon the sorption of ethanol in HFAU at room temperature, followed by heating to 370, 450, 510 and 570 K, are presented in Figure 3A. Upon each heating step, the cell with the zeolite was cooled to room temperature. The spectrum of ethanol physisorbed at room temperature showed the bands of CH_3_ and CH_2_ deformation vibrations at 1395 and 1450 cm^−1^.

Heating to 370 K does not change the spectrum of ethanol, but at 450 K, the bands of ethanol decreased and a new weak and narrow band at 1490 cm^−1^ appeared. This band increased upon heating to 510 K. This band may be attributed to polyethylene [45]. The increase in broad absorption band around 1600–1700 cm^−1^ may be due to water formation. These results can be explained by the dehydration of ethanol on very strong protonic sites, producing water and ethene, which polymerizes, forming polyethylene. Dehydration of ethanol in the presence of acid sites in zeolites ZSM-5 was also reported by Bun et al. [23].

### 2.3. Reaction of Ethanol in Cu(2)HFAU

Zeolite Cu(2)HFAU contains a large amount of Cu^+^_exch_, but much smaller amounts of Cu^2+^ Cu^+^_ox_ are absent. This was evidenced in the experiments of CO and NO sorption (Figure 2B,C), in which an intense 2158 cm^−1^ band of Cu^+^-CO and relatively weak 1850–1900 cm^−1^ band of Cu^2+^-NO as well as the absence of 2130 cm^−1^ band of Cu^+^_ox_-CO were seen.

The spectrum of ethanol sorbed at room temperature in zeolite Cu(2)HFAU is presented in Figure 3B. Heating to 370 K does not change the spectrum; however, heating to temperatures higher than 370 K produces ethene, characterized by the bands at 1428 cm^−1^ (deformation of CH_2_) and a weaker one at 1538 cm^−1^ of C=C stretching. This last band is very weak because the C=C vibration is IR-inactive in free ethene molecules. However, if the ethene molecule is bonded to Cu^+^ ions, this interaction changes the symmetry of the molecule, and the C=C stretching becomes IR-active. The same situation was already observed in our earlier studies of adsorption of ethene [14,46] interacting with Cu^+^ in zeolites CuZSM-5 and CuX. It is interesting to notice that ethene does not polymerize even in the presence of acid sites. It seems that the interaction of π electrons of ethene with Cu^+^ protects the molecule against the attack of protons and formation of carbocation, which is an indispensable step of polymerization. Cu^+^_exch_ does not participate in ethanol oxidation.

### 2.4. Reaction of Ethanol in Cu(5)NaFAU

The Cu sites in zeolite Cu(5)NaFAU are different to those in Cu(2)HFAU. In the case of Cu(2)HFAU, a large amount of Cu^+^_exch_ and a small Cu^2+^ content are present, while for zeolite Cu(5)NaFAU, a much smaller amount of Cu^+^_exch_ and comparable amount of Cu^+^_ox_ content were detected. Additionally, it contains also high amounts of Cu^2+^.

The spectra of ethanol sorbed in Cu(5)NaFAU at room temperature (Figure 3C) is the same as the spectrum of ethanol sorbed in HFAU and Cu(2)HFAU at the same temperature. It shows only the bands of CH_2_ and CH_3_ of ethanol. However, new IR bands appear at 1355, 1630 and 1730 cm^−1^ at higher temperatures. The band at 1630 cm^−1^ is typical of deformation vibration of water molecule, as evidenced by the fact that the same band is present in the spectrum of water sorbed in our zeolite (Figure 3C). The bands at 1355 and 1730 cm^−1^ may be attributed to acetaldehyde, because the same bands are present in the spectrum of acetaldehyde sorbed in the same zeolite (Figure 3C). The bands of ethanol (1395 and 1450 cm^−1^) are not present. The presented result evidence that ethanol was oxidized to acetaldehyde at 450 and 510 K with the formation of water. The spectrum recorded at 450 K also showed the shoulder at ca. 1600 cm^−1^. The heating of the zeolite with reactants (ethanol, acetaldehyde and water) to 510 K caused the decrease in the bands of acetaldehyde with a simultaneous increase in the 1630 cm^−1^ band of water, evidencing the further reaction of acetaldehyde at higher temperature, in which water was produced. The evacuation at 450 K (Figure 3D) caused the decrease of the bands of acetaldehyde and water; the bands at 1405, 1450, 1605, and 1640 cm^−1^ of the surface species, which survived evacuation, were present. The spectrum of these stable surface species showed similar IR bands to the spectrum of coper acetate (Figure 3D), suggesting that some acetaldehyde molecules were oxidized to acetic acid, which reacted with Cu-oxides or with the zeolite framework, forming the acetate (CH_3_COO^-^) species and water. The bands at 1450 and 1605 cm^−1^ may be due to symmetric and antisymmetric vibrations of the COO^-^ species.

The reactions of the oxidation of ethanol on some oxides and noble metals supported on these oxides were studied by several authors [47,48,49,50,51,52]. On CeO_2_ and ZrO_2_, ethanol formed ethoxy species (by the reaction with surface hydroxyls), which were subsequently oxidized to acetaldehyde and to the acetate species; the same reactions were observed for CeO_2_ supported metals. For the CuO/ZrO_2_ system, three kinds of ethoxy species were found [52]: linear ethoxyls (1100 and 1150 cm^−1^ C-O stretching bands) and bridged ethoxyls (1060 cm^−1^). At higher temperatures, the ethoxy species produced acetaldehyde and hydrogen (this was found by mass spectrometry). Acetate forms were also produced (IR bands 1440 and 1550 cm^−1^). An interesting observation was the production of acetone, which was the product of condensation. On some metals, such as Pt or Pd supported on CeO_2_, acetaldehyde was formed, but at higher temperatures, acetaldehyde condensed, forming other organic molecules, such as crotonaldehyde. On the other hand, CO was formed in the presence of Rh, indicating the power of Rh in the breaking of the carbon–carbon bond in acetaldehyde.

It should be noted that in our study, the oxidation of ethanol and acetaldehyde over zeolite Cu(5)NaFAU took place without the supply of molecular oxygen, so the oxygen donors may be the Cu-species present in this zeolite. Zeolite Cu(5)NaFAU contains Cu^+^_ox_ and Cu^2+^_ox_ (CuO). These species may be potential donors of oxygen; however, in the presence of water, Cu^2+^_exch_ may also produce CuO due to hydrolysis and further dehydration of hydroxycations. All these Cu-oxide forms may act as oxygen donors.

### 2.5. Oxygen Donors in Ethanol Oxidation

The information on the role of Cu species in our zeolite Cu(5)NaFAU in ethanol oxidation was obtained in the experiments, in which CO and NO were sorbed before and after the reaction of ethanol at 450 K (Figure 4A,B). At this temperature, ethanol is oxidized to acetaldehyde and further to the acetate species (Figure 3C). The CO sorption experiments evidenced that most of Cu^+^_ox_ (characterized by the Cu^+^_ox_-CO band at 2130 cm^−1^) were consumed by the reaction of ethanol at 450 K. Similarly, the reaction with ethanol at the same temperature reduced the amount of Cu^2+^ (the Cu^2+^-NO bands 1800–1900 cm^−1^ decreased). These results evidence that at 450 K, both Cu^+^_ox_ and Cu^2+^ species may be donors of oxygen in the oxidation of ethanol, producing acetaldehyde undergoing further oxidation toward the acetate species.

Further information on the role of Cu^2+^ and Cu^+^_ox_ in the oxidation reactions was obtained in the experiments, in which ethanol was sorbed in zeolite of different proportions between Cu^2+^ and Cu^+^_ox_ contents. In one series of experiments, zeolite was activated at 720 K, and then oxidized by oxygen treatment at 570 K. In the second one, it was activated at 470 K instead of 720 K. In turn, in the last series, ethanol was sorbed over zeolite on which the oxidation of ethanol was performed prior to the sorption of ethanol.

In the first series of experiments, zeolite Cu(5)NaFAU was oxidized in oxygen at 570 K and next, evacuated at the same temperature. The IR spectra of CO sorbed in this zeolite evidenced a decrease in the band of CO interacting with Cu^+^_ox_ (Figure 5A). The experiments of NO sorption indicated, in an indirect way, the increase in Cu^2+^ content (NO band at 1850–1900 cm^−1^ grows, Figure 5B) compared to non-oxidized zeolite. Therefore, the treatment with oxygen decreased the contribution of Cu^+^_ox_ and increased the amount of Cu^2+^. The spectra recorded upon the sorption of ethanol at room temperature and heating to 450 K are presented in Figure 5C and may be compared with the spectra recorded upon sorption of ethanol in non-oxidized zeolite. The reaction of ethanol in oxidized zeolite produced more acetaldehyde than for the zeolite that was not treated with oxygen. The shoulder at ca. 1600 cm^−1^ was absent. This result suggests that Cu^2+^ species are donors of oxygen in the oxidation of ethanol to acetaldehyde, but for further reaction, i.e., oxidation of acetaldehyde to acetate species, Cu^+^_ox_ forms are necessary.

In the second series of experiments, zeolite Cu(5)NaFAU was activated in vacuum at 470 K instead of 720 K. Zeolite activated at 470 K contains less of both Cu^+^_ox_ and Cu^+^_exch_, as well as somewhat more of Cu^2+^, which was evidenced by low intensities of Cu^+^-CO bands (Figure 5A) and by higher intensity of the Cu^2+^-NO band (Figure 5B), respectively. This observation may be explained by the transformation of CuO into Cu_2_O at higher temperatures. According to the data presented in Figure 5C, the reaction of ethanol over Cu(5)NaFAU activated at 470 K produced more acetaldehyde than over the catalyst activated at 720 K, and did not produce acetate species. It again confirms the importance of Cu^+^_ox_ in the oxidation of acetaldehyde to acetates.

The third series of experiments was the reaction of ethanol over zeolite Cu(5)NaFAU, in which the oxidation of ethanol was already done. In these experiments, ethanol was adsorbed in freshly activated zeolite, and upon heating to 450 K and subsequent evacuation at 570 K, the zeolite was cooled to room temperature. Ethanol was again adsorbed at room temperature, subsequently, and heated to 450 K. The spectrum recorded upon heating to 450 K; subsequent cooling to room temperature (Figure 5C) shows the band of acetaldehyde (1730 cm^−1^), but the shoulder at ca. 1600 cm^−1^, typical of the acetate species, is absent. The experiments of CO and NO sorption (Figure 5A,B) evidenced the decrease in the amount of both Cu^+^_ox_ and Cu^2+^. These results evidence that the oxidation of ethanol and the production of acetaldehyde and the acetate species consumes some Cu^2+^ and most of the Cu^+^_ox_. In turn, the reaction of ethanol on such treated zeolite (Figure 5C) produces acetaldehyde without the acetate species (Figure 5C). The results obtained in all these three series of experiments suggested that Cu^2+^ species (most probable CuO) are oxygen donors in ethanol oxidation to acetaldehyde, and Cu^+^_ox_ species are responsible for acetate formation.

The information on the role of Cu species in ethanol oxidation was also obtained in the experiments, in which ethanol was adsorbed on bulk CuO and Cu_2_O mixed with SiO_2_ and heated to 370, 450, 510 and 570 K.

The information on the presence of Cu sites was obtained by the adsorption of CO and NO as probe molecules (Figure 6). The presence of Cu^+^ and Cu^2+^ on the surfaces of both oxides was found. The concentration of Cu^+^ was higher on Cu_2_O; meanwhile, for CuO, the contribution of Cu^2+^ was dominating.

The spectra recorded upon ethanol adsorption and heating to higher temperatures are presented in Figure 7A,B. The spectra of ethanol recorded at room temperature on both oxides are the same as in the case of ethanol sorbed in zeolites (Figure 3C). Heating to 370 K did not produce acetaldehyde, but small amounts of acetaldehyde were formed at 450 K. In the case of CuO, the amount of acetaldehyde attained a maximum at 510 K; the amount decreased at 570 K due to oxidation of acetaldehyde with the formation of water (band at 1630 cm^−1^), as well as CO and CO_2_ (spectra not shown). The small loss of acetaldehyde is well seen in the difference spectrum (top spectrum in Figure 7A). The situation is different for Cu_2_O (Figure 7B), in which heating to 570 K produced acetic acid. It may be supposed that acetic acid was also formed in the reaction of ethanol on zeolite Cu(5)NaFAU, but acid reacted with zeolite and formed the acetate species. The reaction of acetic acid in zeolites producing acetate ions was already observed [53].

The difference spectrum (between spectra recorded upon heating to 570 and 510 K) is very similar to the spectrum of acetic acid adsorbed on SiO. It should be noted that acetic acid was not formed for CuO (Figure 7A).

Generally, it can be postulated that Cu^2+^ (mostly CuO) is a donor of oxygen in the oxidation of ethanol to acetaldehyde; however, Cu^+^_ox_ is responsible for the oxygen supply in further oxidation of acetaldehyde to acetic acid (or acetate species). Unfortunately, the role of Cu^+^_ox_ in the oxidation of ethanol to acetaldehyde cannot be evidenced due to the impossibility of the investigation of this reaction on materials containing only Cu^+^_ox_. Neither zeolite nor Cu_2_O containing only Cu^+^_ox_ without Cu^2+^ can be obtained. Bulk Cu_2_O always contains some Cu^2+^ produced by the contact with atmospheric air (Figure 6B). The CuO and Cu_2_O particles may be situated inside zeolitic channels and pores or can be present on external surfaces of zeolite crystals.

The role of oxygen-containing Cu species as oxygen donors in the oxidation of benzyl alcohol to benzaldehyde, CO and CO_2_ on Cu-ZSM-5 zeolite was studied by Nakao et al. [24]. Catalytic performance results were completed with the IR studies of CO and benzaldehyde sorption. An interesting observation was the formation of relatively small amounts of benzaldehyde even in the absence of gaseous oxygen, suggesting that oxygen-containing Cu species were a source of oxygen in the reaction. IR spectra of CO sorbed on CuZSM-5 zeolites showed the bands at 2156 and 2135 cm^−1^, which can be assigned to Cu^+^_exch_-CO and Cu^+^_ox_-CO, respectively. The authors indicated that the Cu^2+^ and Cu^2+^-O_2_ adducts reacted with benzyl alcohol, producing benzaldehyde and Cu^+^, which was subsequently reoxidized with molecular oxygen. The results obtained in our study are consistent with the conclusions of Nakao et al. [24].

## 3. Materials and Methods

### 3.1. Materials

Pristine zeolite of faujasite-type structure denoted as HFAU (Si/Al = 31) was produced by Zeolyst company (CBV 760). It was dealuminated by steaming and acid treatment by the producer. Cu-containing zeolites Cu(2)HFAU and Cu(5)NaFAU were obtained by the impregnation method with 0.5 M Cu(NO_3_)_2_ solution. The amounts of Cu ions in solutions corresponded to 2% or 5% of Cu in impregnated zeolites. Zeolite Cu(2)HFAU was obtained by the impregnation of pristine HFAU. It contained 2 wt.% of Cu. In order to obtain Cu(5)NaFAU, zeolite HFAU was transformed into sodium form by four-fold exchange with 0.5 M NaNO_3_, and then washed in distilled water. The resulting NaFAU was subsequently impregnated with 0.5 M Cu(NO_3_)_2_, producing zeolite containing 5 wt.% of Cu. All samples were dried at 390 K and next calcined at 770 K.

Bulk CuO and Cu_2_O (Aldrich) were used in the IR measurements. Both oxides were mixed with SiO_2_ Cabosil in proportion with 8 wt.% of oxides, and 92 wt.% of SiOEthanol, acetaldehyde and acetic acid (Aldrich) were used in IR adsorption experiments.

### 3.2. IR Studies

In most cases, prior to the IR experiments, zeolites were evacuated in situ in an IR cell at 720 K for 1 h. The CuO/SiO_2_ and Cu_2_O/SiO_2_ wafers were evacuated at 470 K and 710 K, respectively, for 1 h. In some experiments, zeolite Cu(5)NaFAU was pretreated in vacuum at 720 K, then subsequently treated with oxygen at 570 K for 1 h and finally, evacuated at the same temperature.

The spectra were recorded with a NICOLET 6700 spectrometer (Thermo Scientific, Cambridge, MA, USA) with the spectral resolution of 1 cm^−^CO, and NO (Air Products) were used as probe molecules. The adsorption of CO was performed at room temperature. Adsorption of NO was done at ca. 190 K.

## 4. Conclusions

The coadsorption of ethanol and CO evidenced the transfer of electrons from ethanol to Cu^+^ and to molecules bonded to the same Cu ion as CO. Ethanol adsorbed on zeolite HFAU underwent dehydration toward ethane, which polymerized. Dehydration of ethanol also took place on zeolite Cu(2)HFAU, containing protonic sites and Cu^+^_exch_, but in this case, ethene was bonded to Cu^+^ and did not polymerize. The C=C stretching, which is IR-inactive in free molecule, became active when ethene interacted with Cu^+^. Ethanol adsorbed on Cu(5)NaFAU containing mostly Cu^+^_ox_, Cu^2+^_exch_ and Cu^2+^_ox_ (CuO) was oxidized to acetaldehyde, and further to the acetate species. As oxygen was not supplied, the donors of oxygen in the considered reaction could be oxides of Cu^+^ and Cu^2+^. In order to learn the role of these copper species in this type of reaction, the CO and NO sorption in Cu-zeolite before and after ethanol oxidation were carried out. Moreover, the oxidation of ethanol was performed on zeolites, in which the proportion between the amounts of Cu^+^_ox_ and Cu^2+^_ox_ was changed by various treatments. Based on all these experiments, as well as the reaction of ethanol over bulk CuO and Cu_2_O, it may be concluded that ethanol was oxidized to acetaldehyde by Cu^2+^_ox_ and probably also by Cu^+^_ox_, whereas Cu^+^_ox_ was responsible for the further oxidation of acetaldehyde to the acetate species.

## Figures and Tables

**Figure 1 molecules-26-02669-f001:**
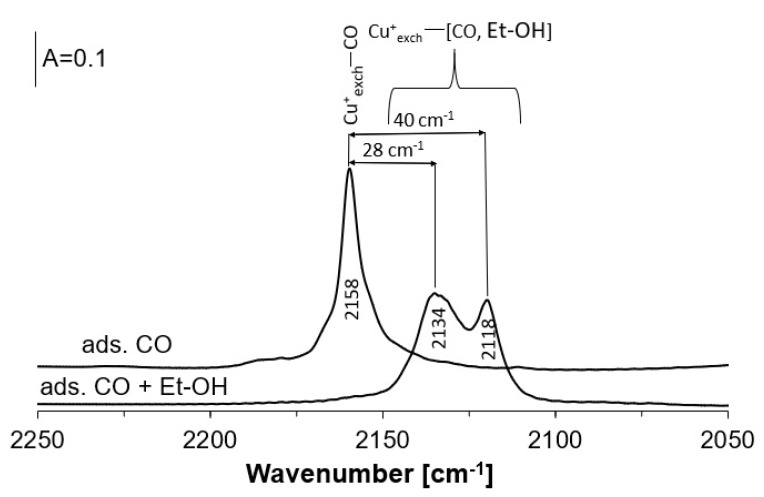
The spectrum of CO sorbed in zeolite Cu(2)HFAU and the same zeolite with preadsorbed ethanol.

**Figure 2 molecules-26-02669-f002:**
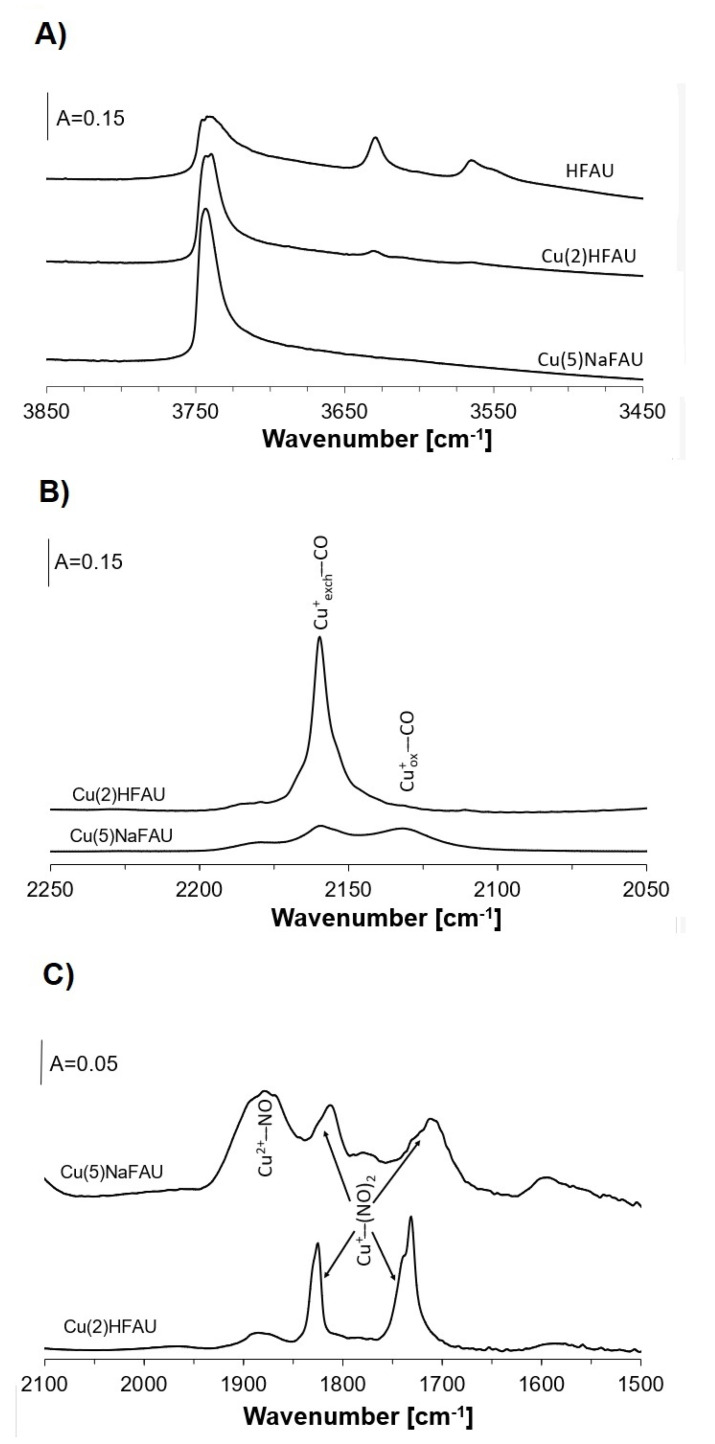
(**A**) The spectra of OH groups in zeolites HFAU, Cu(2)HFAU and Cu(5)NaFAU; B, C– the spectra of CO sorbed at room temperature in zeolites Cu(2)HFAU (**B**) and of NO sorbed in Cu(5)NaFAU at 190 K (**C**).

**Figure 3 molecules-26-02669-f003:**
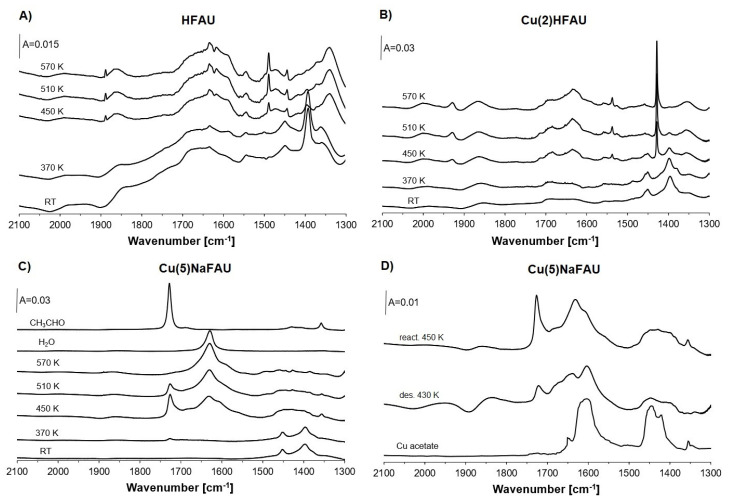
The spectra recorded upon the sorption of ethanol in zeolites HFAU (**A**), Cu(2)HFAU (**B**) and Cu(5)NaFAU (**C**) and heating to 370, 450, 510 and 570 K. The spectra recorded upon the reaction of ethanol in Cu(5)NaFAU at 450 K and evacuation at 430 K are presented in (**D**).

**Figure 4 molecules-26-02669-f004:**
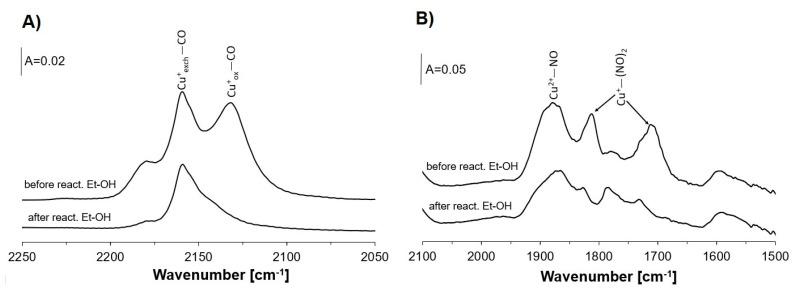
The spectra of CO (**A**) and NO (**B**) sorbed at room temperature (CO) and at 190 K (NO) in zeolite Cu(5)NaFAU before and after the reaction with ethanol at 450 K.

**Figure 5 molecules-26-02669-f005:**
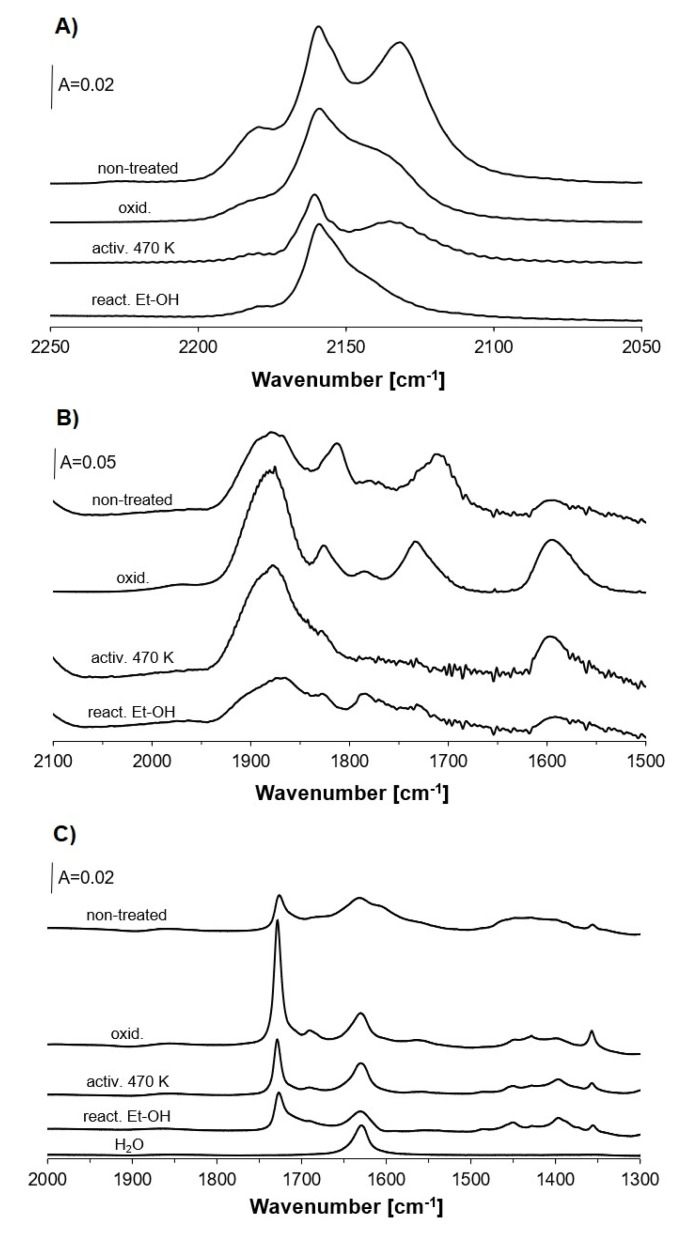
The spectra of CO sorbed at room temperature (**A**), NO sorbed at 190 K (**B**) and the spectra recorded upon sorption of ethanol and heating to 450 K (**C**) in zeolite Cu(5)NaFAU non treated, the same zeolite oxidized, zeolite activated at 470 K as well as zeolite in which the reaction of ethanol was carried out. The spectrum of water sorbed in zeolite is also presented.

**Figure 6 molecules-26-02669-f006:**
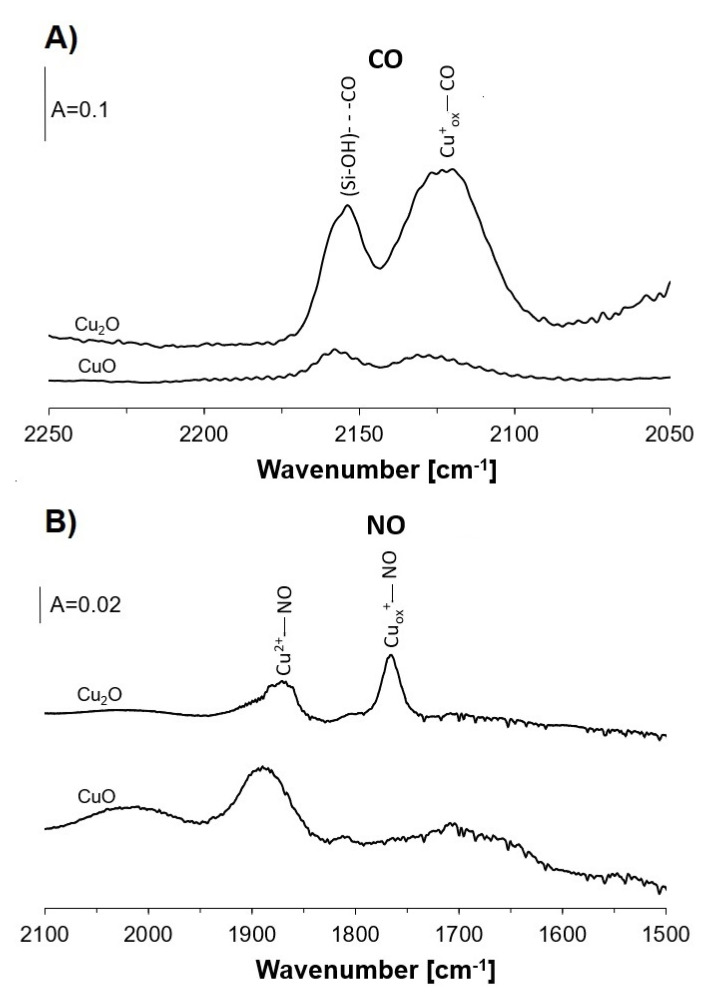
IR spectra of CO (**A**) and NO (**B**) adsorbed at 170 K on Cu_2_O and CuO diluted in SiO_2_.

**Figure 7 molecules-26-02669-f007:**
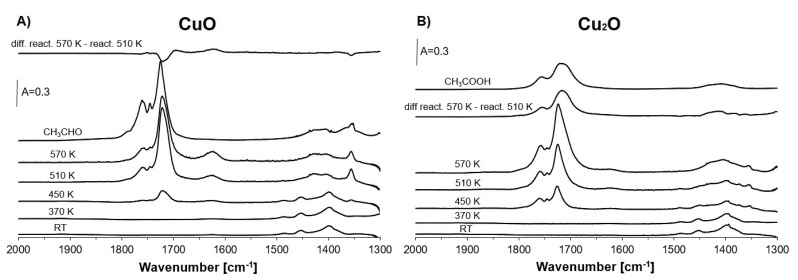
The spectra recorded upon the adsorption of ethanol on CuO/SiO_2_ (**A**) and on Cu_2_O/SiO_2_ (**B**) at room temperature and heating to 370, 450, 510 and 570 K as well as the differences between the spectra recorded upon the heating to 570 K and 510 K. The spectra of acetaldehyde and acetic acid are also presented.

## Data Availability

Not applicable.

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
