# Peer review of "Oxidation of Ethanol in Cu-Faujasites Studied by IR Spectroscopy"

_molecules, 2021, doi:10.3390/molecules26092669_

Round 1

Reviewer 1 Report

see attached

Author Response

REVIEWER 1

We are thankful the  Reviewer for helpful comments.

REMARK 1

The paper is devoted to understand the role of copper cations in various positions in/out FAU zeolites by means of infrared spectroscopy performed with or without co-adsorption of ethanol and CO or NO. The amount of Cu was varied in H-FAU or Na-FAU. The assignments and discussion of results are mostly sound. Revision is asked because of the way the paper is sometimes written, which could profitably be made clearer.

REPLY 1:

Whole text of manuscript has been checked and partially restored. We also corrected and simplified many sentences in order to make the text more friendly to read.   

REMARK 2

(2.4. Reaction of ethanol in Cu(5)NaFAU. This paragraph is very long and it would benefit from restructuring. Several points are exposed: ethanol alone (line 171), digressions about literature (line 192), ads CO and NO (206), pre-treatment with oxygen (217), ethanol + ethanol, and other catalysts Cu/SiO2 (line 265). “Summing out, it can be said that Cu2+ (mostly CuO)…” (line 287): if CuO where these (nano)particles would be or where these clusters would be?

REPLY 2:

a. Actually, the paragraph entitled “2.4 Reaction of ethanol in Cu(5)NaFAU” is long and contains a lot of information. According to reviewer’s suggestion, this section has been divided into two parts: “2.4 Reaction of ethanol in Cu(5)NaFAU” and “2.5 Oxygen donors in ethanol oxidation”.

b. line 287 “Generally, it can be postulated that..” - small CuO clusters may be located inside zeolitic channels (zeolite of faujasite type have very large channels), inside pores produced during dealumination made by producer, or on the outer surfaces of zeolitic crystals. This information has been inserted into the text – lines 311-313.

REMARK 3

Line 39, Introduction, there is no mention of co-adsorption of NO, nor of pre-treatment with O2, nor of ethanol being adsorbed alone, etc., experiments which should be introduced as well.

REPLY 3

We are not sure about the intention of Reviewer, but we understood that more information on the results obtained by Tsuruva (refs 21, 22) is needed. Therefore, we have added more results of conclusions from these references (lines 41-44).

REMARK 4

Lines 68-69, I suggest a bit more explanations about Cu+ “in exchange extra-framework positions”, if the authors wish to reach readers less used to such zeolites. A scheme or drawing would be nice to show all the types of copper species.

REPLY 4:

According to Reviewer advice, we inserted the explanation on “cations in extraframework position” and “cations in oxide form” – lines 74-77.

REMARK 5

3.1 Materials, line 317, “Bulk CuO and Cu2O (Aldrich) were used in IR studies. Both oxides were mixed with SiO2 etc.” There is no other experiment giving some features of these solids? XRD, XPS (e.g. to show the ratio Cu2+/Cu+ in Cu2O)?

REPLY 5:

The Referee pointed out an important problem i.e. the characterization of bulk CuO and Cu2O. The most important property is the presence of Cu2+ and Cu+ on the surfaces of these oxides. It was studied by the adsorption of probe molecules CO (probe for Cu+) and  NO (probe for Cu2+). It was found that both oxides contain Cu+ and Cu2+ in oxide forms on the surfaces. The concentration of Cu+ was higher on Cu2O, and contribution of Cu2+ was higher on CuO. This information was inserted into the text lines 278-281 and Fig. 6 was added.

REMARK 6

Several sentences have to be edited, lines 81, 83, 91 (“also the coadsorption of ethanol and CO molecules on one Cu+ cation.”), 100 (“Cu+exch. “solo” shows”), 108-111, 158 (ethane?), 215-216, 221-222, 342-343. The whole text must be carefully edited.

REPLY 6:

a. According to Referee advice several sentences have been edited - lines 85-87, 102-107, 113-117, 363-366. Furthermore, the names “solo” and “one Cu+ cation” have been removed.

b. Many thanks for careful reading the text and finding an important mistake. Instead of word “ethene”, text editor inserted “ethane”. In most cases, we have corrected the mistake, but in four places we missed the mistake, and “ethane” remained. Thank you again for pointing out the mistake.

Reviewer 2 Report

In my opinion, the work is interesting, well written and can be published in Molecules

Author Response

Thank you for your positive review.

Reviewer 3 Report

#1:  Although the authors show IR spectra "only".  Quantitative and deeper discussion ,for example, using peak area and band shift, etc, is desired to quantitative understanding of molecular dynamics occured within the zeolite cages.

#2:  Abstract:  Ethane do not have C=C bond, so "C=C stretching which is IR non-active in free ethane molecule" is  weird.

#3: "important amounts" is inappropriate. 

Author Response

We are thankful the  Reviewer for helpful comments.

REMARK 1

Although the authors show IR spectra "only".  Quantitative and deeper discussion ,for example, using peak area and band shift, etc, is desired to quantitative understanding of molecular dynamics occured within the zeolite cages.

REPLY 1:

Even though we realized a lot of quantitative IR studies concerning concentration of acid sites in zeolites and also concerning Cu sites in zeolites in the past (e.g. ref. 39), this study has mostly qualitative aspect: we observe that IR bands of ethanol are replaced by those of acetaldehyde, and next by acetate species.

REMARK 2

Abstract:  Ethane do not have C=C bond, so "C=C stretching which is IR non-active in free ethane molecule" is  weird.

REPLY 2:

As answered to Reviewer 1 that the text editor changed the word “ethene” by “ethane” We are thankful the Reviewer 2 for pointing out this mistake. It has been corrected.

REMARK 3

"important amounts" is inappropriate. 

REPLY 3:

According to Reviewer advice, the term “important amounts” has been replaced by “big amounts”. 

Reviewer 4 Report

The article entitled “Oxidation of ethanol in Cu-faujasites studied by IR spectros- 2 copy” is an interesting report on the preparation of Cu-faujasites for the co-adsorption of carbon monoxide and ethanol and subsequent oxidation of the latter. The authors have developed thorough IR adsorption studies at different temperatures for the selected materials. For the sake of comparison, the authors have also studied the same phenomenae on CuO and Cu2O in presence of different gases and at different temperature values. This paper can be granted publication in "Molecules" after some minor changes are completed:

Line 27 – Cu-containing zeolites

Line 88- relatively large amounts

Line 95 – coadsorption

Line 96 – to follow how

Line 103-104 electrons are not transmitted, they are transferred. “result of electron transfer” is the correct expression to use.

Line 114 – Use different colours or line styles for the FTIR spectra.

Line 117 – Reference to Fig 2A is not needed here without further explanation than Brønsted acid sites and of all the relevant content of the figure.

Lines 122-23 – Please show “the bands of C-H stretching (2900 122 – 3000 cm-“ in sup info or remove the fragments. If it’s relevant, it must be shown.

Line 124 – what is “the ethanol spectrum”? What substrate does it refer to? This is confusing.

Line 143 – intense band

Line 146 – Compare with line 124. This is a lot better.

Line 154 – CuX, in which we could confirm that dehydration..

Line 156-160 Where is the evidence or reference for the observation with bu-1-ene, CuZSM-5, etc?

Line 167 – different to those

Line 178 – please consider disappear vs are not present.

Line 180 – “Water molecule was formed too”. What does this exactly mean?

Line 187 – shows similar IR bands to the spectrum of copper acetate

Line 193 – Please give detail of the information present in these references as you discuss your results.

Lines 215-16 – Please translate from Polish to English or remove.

Line 220 – Give detail of this oxygen treatment in Materials and Methods section

Line 229-30 – The increase of this amount is only indirect. Please reword accordingly in all the text.

Line 234-5 – Nonoxidized = reduced? What species are you referring to?

Line 266 – How do you dilute CuO or Cu2O in SiO2? Mixed could be a better wording?

Line 290 – cannot be revealed = is unclear?

Lines 312 and 315 – How did the authors confirm 2% and 5% percentages?

Author Response

We are thankful the  Reviewer for careful reading our text, finding out “weak points” and helpful comments.

REMARKS /REPLIES

The article entitled “Oxidation of ethanol in Cu-faujasites studied by IR spectroscopy” is an interesting report on the preparation of Cu-faujasites for the co-adsorption of carbon monoxide and ethanol and subsequent oxidation of the latter. The authors have developed thorough IR adsorption studies at different temperatures for the selected materials. For the sake of comparison, the authors have also studied the same phenomenae on CuO and Cu2O in presence of different gases and at different temperature values. This paper can be granted publication in "Molecules" after some minor changes are completed:

Line 27 – Cu-containing zeolites: it has been corrected (line 28).

Line 88- relatively large amounts: it has been corrected (line 94).

Line 95 – coadsorption it has been corrected (line 101).

Line 96 – to follow how has been changed to “to learn how” (line 102).

Line 103-104 electrons are not transmitted, they are transferred. “result of electron transfer” is the correct expression to use. “the result of electron transfer” has been used (lines 109-110).

Line 114 – Use different colours or line styles for the FTIR spectra. In our opinion and other reviewers, the IR spectra are well described (captions are clear), and we think that there is no need to change colors/styles in the given pictures.

Line 117 – Reference to Fig 2A is not needed here without further explanation than Brønsted acid sites and of all the relevant content of the figure.

This reference has been removed.

Lines 122-23 – Please show “the bands of C-H stretching (2900 122 – 3000 cm-“ in sup info or remove the fragments. If it’s relevant, it must be shown.

This information has been removed.

Line 124 – what is “the ethanol spectrum”? What substrate does it refer to? This is confusing. This information has been removed.

Line 143 – intense band

This information has been modified. (line 144)

Line 146 – Compare with line 124. This is a lot better.

The following fragment has been introduced: “The spectrum of ethanol sorbed at room temperature in zeolite Cu(2)HFAU is presented in Fig. 3B. The heating to 370 K does not change the spectrum,  however, the heating to temperatures higher than 370 K produces ethene characterized by the bands at 1428 cm-1 (deformation of CH2) and weaker one at 1538 cm-1 of C=C stretching.” (lines 146-149)

Line 154 – CuX, in which we could confirm that dehydration.

In earlier study we studied the interaction of ethene with Cu+ sites, but ethene was not produced by dehydration of ethanol, gaseous ethene was contacted with zeolite.

Line 156-160 Where is the evidence or reference for the observation with bu-1-ene, CuZSM-5, etc?

This fragment of text has been removed.

Line 167 – different to those

This fragment has been modified. (line 169)

Line 178 – please consider disappear vs are not present.

This fragment has been modified. (line 181)

Line 180 – “Water molecule was formed too”. What does this exactly mean?

This fragment has been changed to: “The presented results evidence that ethanol was oxidized to acetaldehyde at 450 and 510 K with the formation of water.” (line 181-182)

Line 187 – shows similar IR bands to the spectrum of copper acetate

This fragment has been modified. (line 189-190)

Line 193 – Please give detail of the information present in these references as you discuss your results.

The following fragment has been introduced: “For CuO/ZrO2 system three kinds of ethoxy species were found [53]: linear ethoxyls (1100 and 1150 cm-1 C-O stretching bands) and bridged ethoxyls (1060 cm-1). At higher temperatures ethoxy species produced acetaldehyde and hydrogen (this was found by mass spectrometry). Acetate forms were also produced (IR bands 1440 and 1550 cm-1). An interesting observation was the production of acetone which was the product of condensation.” (lines 198-203)

Lines 215-16 – Please translate from Polish to English or remove.

This sentence has been removed.

Line 220 – Give detail of this oxygen treatment in Materials and Methods section

The following fragment has been introduced: “In some experiments, zeolite Cu(5)NaFAU was pretreated in vacuum at 720K and subsequently, it was treated with oxygen at 570K for 1 h and finally evacuated at the same temperature.” (lines 343-345)

Line 229-30 – The increase of this amount is only indirect. Please reword accordingly in all the text.

This fragment has been modified to: “The experiments of NO sorption indicated in indirect way the increase of Cu2+ content (NO band at 1850-1900 cm-1 grows - Fig. 5 B) compared to non-oxidized zeolite.” (lines 236-238)

Line 234-5 – Nonoxidized = reduced? What species are you referring to?

This fragment has been modified to: “than for zeolite which was non treated with oxygen” (line 243).

Line 266 – How do you dilute CuO or Cu2O in SiO2? Mixed could be a better wording?

This fragment has been modified to:  “bulk CuO and Cu2O mixed with SiO2” (line 276).

Line 290 – cannot be revealed = is unclear?

This fragment has been modified to “cannot be evidenced”. (line 308)

Lines 312 and 315 – How did the authors confirm 2% and 5% percentages?

The following fragment has been introduced: “The amounts of Cu ions in solutions corresponded to 2 or 5% of Cu in impregnated zeolites.” (lines 330-331)

Round 2

Reviewer 3 Report

 The reviewer confirmed the author's reply.